# Interventions for Picky Eaters among Typically Developed Children—A Scoping Review

**DOI:** 10.3390/nu15010242

**Published:** 2023-01-03

**Authors:** Mohd Shah Kamarudin, Mohd Razif Shahril, Hasnah Haron, Masne Kadar, Nik Shanita Safii, Nur Hana Hamzaid

**Affiliations:** 1Center for Rehabilitation & Special Needs Studies (iCaRehab), Faculty of Health Sciences, National University of Malaysia, Kuala Lumpur 56000, Malaysia; 2Center for Healthy Ageing & Wellness (H-CARE), Faculty of Health Sciences, National University of Malaysia, Kuala Lumpur 56000, Malaysia; 3Center for Community Health Studies (ReaCH), Faculty of Health Sciences, National University of Malaysia, Kuala Lumpur 56000, Malaysia

**Keywords:** picky eaters, intervention, sensory, nutrition, parenting, eating behaviour

## Abstract

Picky eating in children is often a major source of concern for many parents and caregivers. Picky eaters (PEs) consume limited foods, demonstrate food aversion, and have a limited food repertoire, which hinders their growth and health. These behaviours are common in children with special health care needs despite the rise in typically developing children. This leads to less attention being given to intervention programmes for typically developing children. Therefore, this scoping review aims to investigate the key concept of an existing intervention programme for PE among typically developing children, primarily on the types and approaches selected. A thorough literature search was conducted on three primary databases (PubMed, Emerald In-sight, and Web of Science) using predefined keywords. The literature was then appraised using the Joanna Briggs Institute’s guidelines and protocols, and the PRISMScR checklist. Inclusion and exclusion criteria were also specified in the screening procedure. Results showed that the majority of the interventions in these studies were single-component interventions, with the sensory approach being the type that was most frequently utilised, followed by the nutrition approach and parenting approach. Single and multiple intervention components improved the assessed outcome, with a note that other components may or may not show a similar outcome, as they were not assessed in the single-component intervention. Given the evidence that picky eating is influenced by various factors, a multi-component intervention can provide a substantial impact on future programmes. In addition, defining picky eaters using standardised tools is also essential for a more inclusive subject selection.

## 1. Introduction

Feeding difficulty in children is a primary concern for parents, especially when it can affect their growth [1]. According to a study by Grey et al. [2], childhood malnutrition and growth impairment can increase the likelihood of noncommunicable diseases occurrence later in adult life. Kerzner [3] suggested these children should initially be examined for red flags such as dysphagia, uncoordinated swallowing that results in coughing or choking, recurrent pneumonia, crying (pain) while eating, vomiting, diarrhea, dermatitis, failure to thrive, and aberrant development, such as preterm and autism. If the children’s condition persists after receiving treatment, an examination of feeding methods, parent–child interaction during meals, and behavioural difficulties during meals would be conducted to determine further resolutions. It is only then that these children can be categorised as picky eaters. However, while researchers have utilised a variety of definitions of picky eaters (PEs) to describe the condition [4,5,6], most researchers agree that there is a decreased intake of food diversity, hesitation in trying new foods (food neophobia), and refusal to eat at regularly eating items. These things will interfere with the children’s daily routine and impact the parent–child relationship [7].

The prevalence of PE preschool children varies by country, ranging from 25% to 53% [5,6,8,9]. In Malaysia, few prevalence studies had been undertaken, with a majority being limited to specific places and subjects. According to one study conducted in Kuala Lumpur, 53% of children aged five to ten years were PEs [8], while another study conducted in Kuala Selangor found that 31.8% of preschool children were PEs. It should be noted that the methods for measuring and defining PEs used in these studies may have differed. In general, there are three primary methods for identifying PEs in children [10]. These strategies involve extracting information from the child’s parent or guardian. The first approach is to ask close-ended questions (yes/no response), i.e., “Is your child a PE?” [6,11,12]. The second method is identical to the first, except that it includes more response options (other than yes/no) and occasionally comprises marks to distinguish PE children [4,13,14]. Thirdly, validated questionnaires such as the Child Eating Behavior Questionnaire (CEBQ) [15], the Stanford Feeding Questionnaire (SFQ) [16], the Oregon Research Institute Child Eating Behavior Inventory (ORI CEBI), the Child Feeding Questionnaire (CFQ) [17], and the Toddler Parent Mealtime Behavior Questionnaire (TPMBQ) can be used [18]. Each of these questionnaires examines the child’s PE status but they vary in terms of the items and the format of the score.

Numerous other studies have revealed that PE children can also impact their nutritional status [19,20,21]. PE children are more likely to be underweight and shorter in stature than non-PE children [14,22,23]. The cause of malnutrition is most likely owing to a deficiency in protein, vegetable intake, and overall calorie intake, all of which affect children’s growth [24]. Additionally, a study in Tehran discovered that stunted toddlers consumed less milk and dairy products, nuts, and dried fruits. These foods provide nutrients (calcium, protein, vitamins, and minerals) necessary for children’s growth, particularly for height gain [25]. PE behaviour was also observed to have a strong impact on children’s nutritional status. A long-term study indicated that pre-schoolers who were persistently picky eaters had lower weight and height by the age of 15 compared to non-PE children [26]. PE behaviour not only affects nutritional health but is also linked to micronutrients, especially zinc [27]. This micronutrient is essential for the immune system’s health and as an enzyme cofactor in the body’s metabolic process [28]. This micronutrient shortage contributes to health issues, that include growth issues, and is linked to the nutritional status of children [29].

Most children do not inherit PE behaviours [30]. Cognitive, as well as social and environmental factors, are two crucial things that affect PE behaviour. Cognitive is a term that encompasses sensory perception, categorisation, feelings, and emotions, while the antecedents, the postnatal environment, and the social environment are more linked to social and environmental factors [10]. These factors have been incorporated into intervention programmes aimed at assisting PE children. Using strategies, such as repeated exposure (taste and texture) and multisensory plays, can help change children’s taste preferences and encourage them to like certain foods more [31,32,33,34,35,36,37,38]. In addition, social and environmental factors are incorporated into PE children’s intervention programs by examining the role of parents, peers, and social interaction [39,40,41,42]. The outcomes and types of intervention in PE children’s intervention programmes are not clearly defined, despite the incorporation of various factors and methods. There has not been a study that compiles all forms of intervention programme factors and examines the effects and interactions of each factor on typically developing PE children. Therefore, this scoping review aimed to investigate the key concept of an existing intervention programme for PE among developing children based on the predominant type of intervention and approach.

## 2. Materials and Methods

A search on three (online) databases was conducted according to the Joanna Briggs Institute’s guidelines and protocols [43], the PRISMA-ScR checklist [44], and the Arksey and O’Malley [45] study framework.

### 2.1. Stage 1—Identify the Research Question

Scoping review questions.

What are the key concepts (focusing on the types of intervention and the most common approaches) applied in intervention programs for PE children?What are the reported outcomes of the interventions?

### 2.2. Stage 2—Identify Relevant Studies

Between 21 March and 25 March 2022, a search for related articles using the keywords “(children OR preschool OR toddler) AND (“picky eat*” OR picky OR “fussy eat*” OR fussy* OR neophobia) AND (nutrition OR “healthy eating” OR behaviour* OR sensory*)” was conducted in three databases: PubMed, Emerald Insight, and Web of Science.

### 2.3. Stage 3—Study Selection

The followings were the inclusion criteria.

3.Intervention studies.4.Subjects are preschool children aged between three to five years old.5.Study outcomes focusing on sensory, nutrition, and behavioural aspects associated with picky eating

The following are the exclusion criteria.

6.Studies involving children with non-typical developmental issues.

Two reviewers (M.S.K. and N.H.H) independently screened the articles and the results were compared during a single discussion session. At times when a discrepancy occurs, a third reviewer (M.R.S) would be requested for consultation.

### 2.4. Stage 4—Charting the Data

Following the research questions, full articles were retrieved from pertinent sources and analysed. Two evaluators (M.S.K and N.H.H) independently reviewed each article and documented information such as the author(s), publication year, study location, research objective(s), methodology, subjects, PE screening tools, study type, intervention and delivery details, and research findings.

### 2.5. Stage 5—Collating, Summarising, and Reporting Results

All the identified articles were collected, examined, and reported according to the following themes: (1) screening tools, (2) the type of intervention and the components approach, and (3) intervention outcome and general findings. These data are discussed in the results section of this review.

## 3. Results

### 3.1. Study Characteristic

Following the literature search, a total of 1294 papers were identified (406 PubMed, 247 Emerald Insight, 641 WoS). After removing duplicate articles (n = 332), 962 articles were screened using the inclusion and exclusion criteria of the titles and abstracts, which reduced the total to 16 articles (Figure 1).

This comprehensive scoping review largely included previous studies on intervention programmes conducted for developing children aged three to five years (Table 1), in addition to infants as early as four months old [31,41]. Mallan et al. [41] is a longitudinal study that began when subjects were four months old and continued until they were 3.7 years old, while subjects in Caton et al. [31] ranged in age from 4 to 38 months. Some of the subjects of both studies were within three to five years; therefore, both were considered in this study. There was a total of 2942 participants in these 16 intervention studies. Almost all the identified studies included a control group, except for Caton et al. [31] and Nekitsing et al. [35]. The shortest intervention duration for any given study was one day [32], while the longest was fifteen months [42]. The duration of the interventions was determined by the types of interventions and the objectives of each study. Some studies additionally included post-intervention follow-up sessions. The follow-up period ranged from two to thirty months after the conclusion of the trial [39,40,41,46].

### 3.2. Screening Tools to Identify PE

It was discovered that the definitions of PE used were inconsistent in the selected literature, which is reflected in the different screening tools used (Table 2). It was found that the simplest way to identify PE children was reported in the criteria listed by Ghosh et al. [49] and Khanna et al. [50], such as lack of food diversity, unwillingness to try new foods, lack of interest in eating, hate of specific food groups, etc., in which children were considered a PE when they met at least two of the criteria listed. Three studies relied solely on parental reports of a child’s PE behaviour [36,46,47,48].

Eleven studies did not employ any types of assessments of PEs at the beginning of the trial. However, PE evaluation was used as a confounding factor or an outcome measure in eight of the eleven studies to assess the efficacy of the interventions. The instrument used for the assessment was the Child Eating Behavior Questionnaire (CEBQ) [31,37,39,41,42], the modified version of Carruth, et al. [46,51], the Lifestyle Behaviour Checklist (LBC) [39] and the Child Food Neophobia Scale (CFNS) [32,33,37,42], which were carried out by parents of the subjects.

### 3.3. Types of Intervention and Component Approaches

As shown in Table 2, there are two types of interventions found: single-component and multi-component interventions. A total of thirteen studies used single-component intervention [31,32,33,34,35,36,37,39,46,47,48,49,50], while the remaining three adopted multi-component intervention [40,41,42].

The component approaches used in the 13 single-component studies were sensory, nutrition, and parenting. The sensory component approach (54%) [31,32,33,34,35,36,37] is the most common approach among the 13 studies, followed by the nutrition component approach (38%) [46,47,48,49,50] and the parenting component approach (8%) [39]. The sensory component provides the focal point for repeated exposure and sensory play [31,32,33,34,35,36,37]. In the sensory play, children are allowed to touch, observe, and taste the foods presented as they please. As for the nutrition component approach, it places a primary emphasis on supplementary foods (oral nutrition supplementation or herbal supplementation that aids in enhancing appetite and general well-being), and/or a supplement that was taken with the intention to increase total calories consumed by the children [46,47,48,49,50]. The third component approach is the parenting approach, which consists of general parenting skills, including setting boundaries and providing encouragement. This is commonly used to establish mealtime routines and monitor screen time [39].

Only three studies on multi-component intervention were discovered by the literature search. The cognitive behaviour approach, sensory approach, and social and environmental approach were utilised in the study conducted by Bellows et al. [40]. The cognitive behaviour component employs vegetable cartoon characters in activities such as puppet performances, puzzles, and activity books. In addition, children were repeatedly exposed to targeted vegetables (the sensory approach), and school-wide posters and banners carried a specified message (the social and environmental approach). In contrast, during the intervention phase, the parents received newsletters that contained recipes, spatulas, and chef’s caps to encourage them to cook meals containing the targeted vegetables. The research conducted by Mallan et al. [41] consisted of three component approaches as well, which were the nutrition (food intake) approach, the social and environmental approach, and the parenting approach. In the study, the nutrition approach focused on offering food variety (texture and taste) and repeated neutral exposure to healthy foods. Concurrently, a pleasant feeding environment was fostered, and children were encouraged to eat with their families in more extensive social settings. In addition to this, parents were trained on how to practice an authoritative parenting style. The third study by Skouteris et al. [42] used a multi-component intervention, a two-component approach composed of the parenting approach and the nutrition approach. This study emphasises healthy snacks, cooking together, and portion control for the nutrition approach. At the same time, the parenting approach is based on a model of parenting behaviour.

Regarding implementation, most interventions consist of group activities (discussions, interactive activities, and talks) utilising a standard module. Examples of activities are puppet shows, role-playing, puzzles, reading a storybook, playing with food, and repeatedly exposing the child to the same food.

### 3.4. Intervention Outcomes and General Findings

Depending on the component approach, the outcome to demonstrate the efficacy of a single-component intervention will vary. Behaviour assessment is used to evaluate the effectiveness of the sensory component approach and the parenting component approach (a total of eight studies). Five of the eight studies investigated the acceptance of food through liking/willingness to try (the sensory component approach) to demonstrate the intervention’s effectiveness [31,32,33,34,37]. It was found that all of these studies were able to improve the subjects’ liking/willingness to try the targeted foods. Two of the eight studies evaluated children’s PE behaviour (one used the sensory component approach, and one used the parenting component approach). Only the intervention with the sensory component approach reduced the PE behaviour score [36], while the intervention with the parenting component approach showed no significant difference [39]. Another study tested children’s consumption of vegetables to demonstrate the intervention’s effectiveness (the sensory component approach). The study concluded that it was effective in improving vegetable consumption [35].

The effectiveness of a nutrition component approach is measured by nutrition status and food intake outcome. All five studies measured nutritional status (anthropometry) [46,47,48,49,50], and only two of these five studies examined food intake [46,48]. Four out of five studies on the nutrition component approach used supplemental foods (milk) to boost total calorie intake and food quality [47,48,49,50]. In contrast, one study employed herbal supplements that could boost appetite and well-being [46]. Four interventions improved at least one nutritional status indicator (anthropometry) post-intervention, as determined by the study’s findings [47,48,49,50], while one study did not indicate any significant results [46]. Regarding the assessment of food intake, it was determined that two studies showed an increase in consumption of general calories, and macro and micronutrients, including carbohydrates, vitamins A, C, D, and E, Thiamine, calcium, iron, and zinc [46,48].

The study by Bellows et al. [40] examined the effectiveness of a multi-component intervention by assessing vegetable liking and PE behaviour among children. This study employs three component approaches: nutrition, sensory, and social and the environment. This study indicated that the intervention group preferred the targeted vegetables compared to the control group. The number of vegetables consumed by children in the intervention group who described the targeted vegetable as “yummy” and “just okay” had increased at completion and persisted throughout the follow-up period. In contrast, children in the control group who rated the targeted vegetable as “yummy” did not necessarily consume more. Mallan et al. [41] analysed the efficacy of their research intervention by analysing the consumption of targeted food, food fussiness score, and nutritional status improvements (BMI-for-age). The study showed higher exposure to targeted food among children at 14 months of age increased at 3.7 years. It was also discovered that a high intake of vegetables at 14 months old was associated with a low level of food picky eating. For nutritional status, there was no relationship between the target food consumption at 14 months and the BMI-for-age at the age of 3.7 years. Skouteris et al. [42] evaluated the effectiveness of the intervention based on targeted vegetable consumption, child eating behaviour, and food neophobia. They used a two-component strategy that included nutrition and parenting approaches. The study indicated that the consumption of targeted vegetables rose compared to the control group. Furthermore, children were found to be more sensitive to satiety cues, and the food neophobia score decreased after the intervention was completed compared to the control group.

## 4. Discussion

This scoping review aimed to determine the key concept of an existing intervention programme for picky eaters (Pes) among typically developing children based on the predominant type of intervention and approach chosen. In addition, it is focused on identifying the types of screening tools utilised to identify children who were Pes.

The primary findings established in this study are related to the types of interventions, component approaches, and outcomes. It was discovered that the majority of the studies had considered only one single-component intervention, either the sensory approach (single/double or multisensory), nutrition approach (anthropometry, food intake or behaviour), or parenting approach [31,32,33,34,35,36,37,39,46,47,48,49,50]. The selection of intervention type is mainly determined by underlying causes [10,52] and the primary effect on PE behaviour. According to Chilman et al. [52], sensory aspects are one of the inherent factors that cause PE behaviour (sensitivity to food taste and texture). The sensory component intervention focuses primarily on repeated exposure and sensory play [31,32,33,34,35,36,37]. The sensory component intervention has been demonstrated to increase the behaviour assessment used as an outcome measure. Behaviour assessments include acceptability/liking/willingness to try some food types [33,37] and changes in PE behaviour. As an example, eight to ten exposures to the same food can improve acceptance of the particular food [53]. It was determined that this strategy is effective for PE children who consume a limited food variety [54]. However, parents must be cautious when introducing children to new foods. The nutrient content, quality, and type of food (nutrition aspect) should also be highlighted because optimum growth in children requires specific nutrients, including certain vitamins and minerals [55].

A nutritional component is recognised as one of the effects of long-term PE behaviour, but the sensory component is more closely tied to its underlying reasons [10]. It was found that PEs contribute to the prevalence of wasted/underweight status among children [19,21]. Therefore, it is common to see the use of nutritional components, such as weight and height, being part of an intervention outcome measure for PE children. It has also been proven that the use of supplementary foods is highly effective in enhancing the nutritional status of children when used for a short duration (approximately three months). Although a child’s nutritional status can be improved, the underlying cause of PE behaviour issues that are not commonly addressed can result in further deterioration of nutritional status in the long run. This has been confirmed by Grulichova et al. [26], who found that being a PE in childhood affects weight and height at age 15. A zinc deficiency was also observed to be connected with PE behaviour [27].

The parenting approach is also one factor found to influence PE behaviour [52]. One of the studies focused on fostering an environment that can encourage healthy eating and physical activity [39]. Sandvik et al. [39] demonstrate how parents can support healthy eating in children. Parents who are extremely strict in enforcing rules, utilising the concept of punishment, and adopting one-way communication with their children, were found to correlate positively with PE behaviour. Therefore, parents are highly recommended to communicate more openly, establish clear rules and expectations, and engage in a more problem-solving manner with their children [56,57]. Furthermore, children are likely to be exposed to a wider range of foods if their parents are knowledgeable and constantly provide social and physical support [58]. However, research by Sandvik et al. [39] has observed that interventions addressing parenting components alone are insufficient to reduce PE behaviour.

In addition to the types of interventions, suitable assessments or screening tools are required to assess their effectiveness. Typically, suitable assessments or screenings for PE children are determined by the definition used in studies. Therefore, it is of the utmost importance to obtain a standardised taxonomy that encompasses every factor influencing the behaviour of PEs. This will then establish the intervention components that are more precise and effective in addressing the problem. Five studies utilised several different sets of questionnaires in identifying PE characteristics, mainly focusing on behaviour at mealtimes [46,47,48,49,50]. From this literature, children were mainly classified as a PE when they demonstrated at least two characteristics listed in the chosen questionnaires. This approach relies on the researcher’s understanding of the definition of a PE, which differs considerably between research. The frequently cited characteristics of PEs are lack of food diversity, unwillingness to try new foods and familiar food, a lack of interest in eating, and hate of specific food groups. Validated questionnaires, such as the CEBQ, are another method for evaluating PE behaviour. Children who score above three on the food fussiness sub-scale of the CEBQ are considered to have the characteristics of PEs [59]. Not only does it recognise the behaviour of not eating a new/unfamiliar food, but it also identifies the behaviour of not eating familiar food, which is compatible with the definition used by Taylor, et al. [7]. In addition to the food fussiness subscale, the CEBQ consists of seven other subscales (satiety responsiveness, slowness in eating, food responsiveness, enjoyment of food, desire to drink, emotional overeating, and emotional undereating) that provide a comprehensive explanation of child’s eating behaviour [15]. However, this 35-item questionnaire may take parents more time to complete. Even though only one paper was discovered using CEBQ [39], the literature indicates that it was the most used screening tool [10,52]. All of these evaluations, however, had been conducted by parents, as children were unable to provide accurate responses regarding their eating behaviour. Therefore, this evaluation relies on the judgment and understanding of the parents, which may be biased and subjective. The derived data precision is uncertain and cannot be guaranteed.

In addition to the CEBQ, other validated questionnaires, such as the CFNS, werealso used to detect PE behaviour [32,33,37,39,42]. The use of the CFNS is determined by the definition that the researcher has chosen. If food neophobia is recognised as one of the criteria for PEs, it is acceptable to use the CFNS. The CFNS emphasises the fear of trying new foods [60]. It is comprised of ten basic questions that do not require much time to answer. The LBC has also been used to detect PE behaviour [39]. It has 25 items, five of which are particular to PE behaviour. The five questions analyse children’s responses during mealtime, including whining, shouting, arguing, and tantrums. In addition, it determines whether children consume specific types of foods [61]. However, this questionnaire is designed to assess the lifestyle-related issues of obese children. Due to the diversity of nutritional status among PE children, this questionnaire may not be acceptable for some subjects.

Eleven studies identified in this scoping review do not conduct a screening process for PE behaviour [31,37,41,42]. This is commonly applied in studies that aim to improve children’s acceptance of a particular food group, such as vegetables and fruits. It is well known that all children, regardless of their PE status, avoid this food group [62,63]. In certain research, PE screening is unnecessary; instead, PE assessment is performed after the intervention programme has been completed. These evaluations were provided by parents; therefore, parents must thoroughly understand their child’s behaviour to conduct an appropriate assessment.

Even though a single-component intervention is often used, the problem of PEs is significantly complicated and influenced by several other factors [52,64]. It may be beneficial if the causative factors are identified beforehand and the intervention chosen is tailored to the cause of the PE. If the problem of PEs is caused by children’s sensitivity to the taste and texture of food, then the sensory component approach is the most appropriate choice. In this review, it has demonstrated that it is possible to successfully increase children’s willingness to try new foods, to influence their food preference, and decrease picky eating behaviour. Furthermore, if PE behaviour produces issues with nutritional status and nutrient intake, then the nutritional component approach must be implemented. This component also reports a high degree of success in assisting in the improvement of children’s nutritional status. Single intervention components may be able to fix the measurable outcome, but they may not be able to resolve the PE issue completely. According to the findings, other component approaches lacked sufficient evidence in demonstrating their effectiveness to reduce PE behaviour.

Consequently, a multi-component intervention approach may be more appropriate and effective in assisting children with PE issues [54,65]. Overall, studies that employed a multi-component intervention demonstrated an improvement in the measured outcome (liking and intake of targeted food and PE behaviour) following the conclusion of the intervention period. The problem of PEs is highly complex with varying definitions across multiple studies Therefore, it is not surprising that the evaluation of each study’s outcome varies. Single-component interventions are the most popular type of intervention; however, each single-component intervention study uses a different component approach and outcome evaluation. All studies using a multi-component intervention demonstrated positive effects on the outcome being measured. With only three studies, it is difficult to conclude that multiple-component interventions are more effective than single-component interventions.

Despite the fact that it is challenging to draw conclusions from the current studies, integrating each component of PEs may be beneficial for children. The problem of picky eaters is not created by a single clear cause, but rather by a dynamic interplay between the internal and external influences of children that result in this behaviour. [52]. Therefore, we would suggest an intervention with a multi-component intervention including nutrition, sensory, parenting, and social and environmental components similar to Figure 2. It is believed that this approach can assist to enhance the nutritional status of PE children over time. Each component of the intervention is administered in a distinct method or format. However, most interventions rely on group activities (discussions, interactive activities, and talks) as their primary delivery method. Children’s activities incorporate more creative approaches, such as puzzles, puppets, visual cards, storybooks, music, acting, and mascots. These activities have been found to be engaging for youngsters and simultaneously boost the intervention’s efficacy. More serious formats such as talks, role plays, group discussions, and cooking demonstrations should be utilised for activities involving parents.

In terms of age, the intervention in this study was limited to children aged three to five. However, this age range is not uniform across all studies, and studies will be accepted if they fall slightly outside this range. The age range is determined because the intervention implementation is distinct from the age range. Furthermore, only four component approaches; sensory, nutritional, parenting, and social and environmental are the focus of this study. This study did not consider other elements, such as prenatal [10] or children’s personalities [52]. In addition, this review accepted only English-language studies, which may have limited the findings regarding interventions specific to cultures and ethnicities.

## 5. Conclusions

This scoping review examined 16 articles that investigated various forms of interventions to improve children’s eating habits. It was found that PE children with typical development received less attention despite the condition may lead to more severe health consequences. Most previous reviews concentrated on children with special needs, such as autism, ARFID, down syndrome, etc. It is believed that this is the first review to focus on this type of population regarding the type of intervention and component approach.

This review also discovered that a single-component intervention is the most common intervention, with sensory being the most studied component. This factor is possibly the most common cause of children developing as PEs. In terms of intervention efficacy, single-component and multi-component interventions demonstrated an improvement in the assessed outcome. It is preferable to identify the primary reason for PE behaviour, as this will suggest the necessary components for any intervention programmes in the future. The sensory component approach is for children who are sensitive to the taste and texture of food, while the nutrition component approach is for PE children with growth difficulties. However, this does not guarantee the solving of PE behaviour problems. The issue of picky eaters is not the result of a single apparent cause, but rather a dynamic interaction between the internal and external factors of children that result in this behaviour. Therefore, multi-component interventions are proposed so that improvements made to each aspect can reinforce one another in decreasing PE behaviour.

Obtaining a standard taxonomy of PEs that includes all elements influencing this behaviour is essential. This will result in the establishment of standard screening instruments and the production of precise and efficient PE intervention components. Although some studies did not screen for PE behaviours, it was discovered that this type of intervention was aimed at helping to enhance the acceptance of certain food groups. It indirectly benefits PE children who are often lacking in variety in their food intake.

## Figures and Tables

**Figure 1 nutrients-15-00242-f001:**
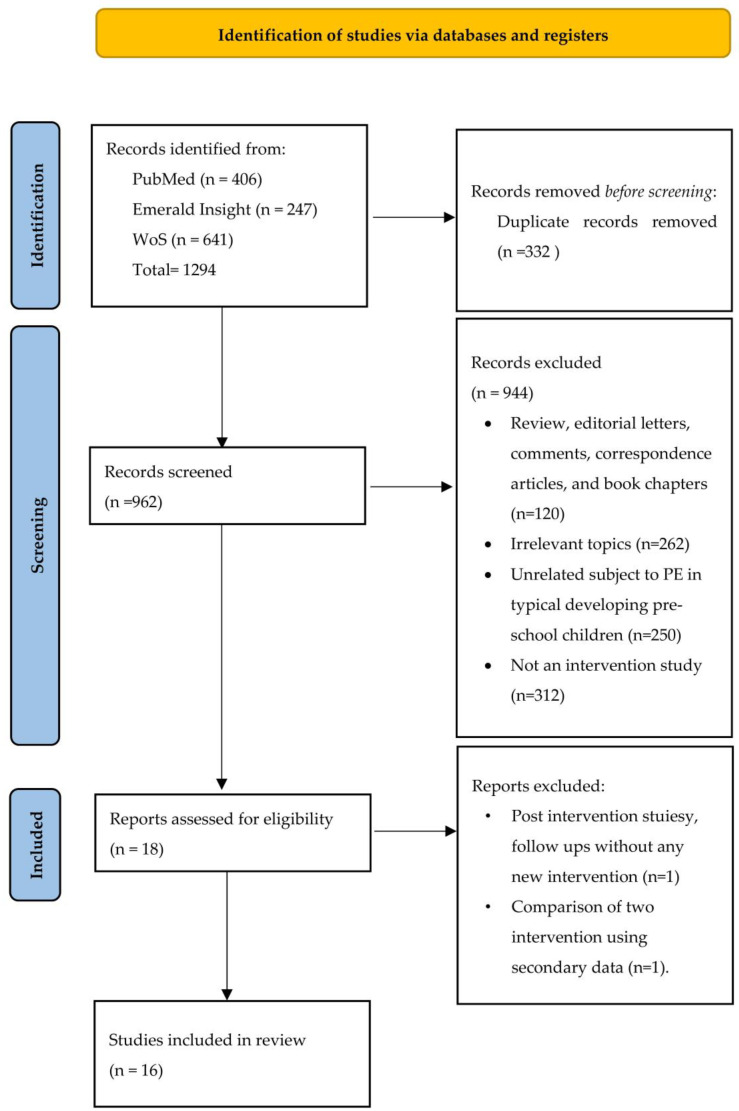
Literature screening process.

**Figure 2 nutrients-15-00242-f002:**
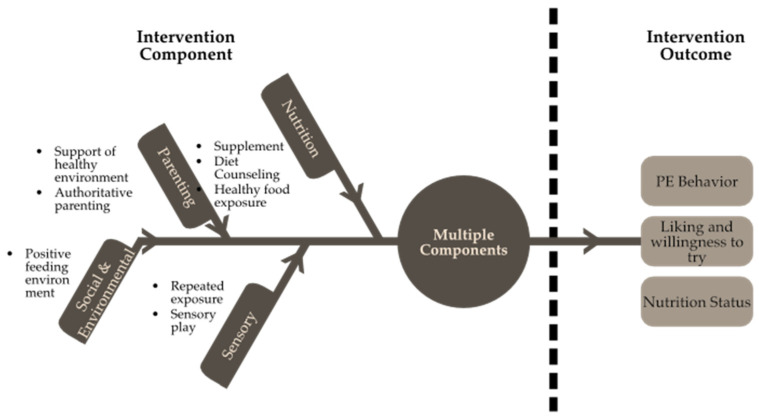
Multi-component intervention in PE children.

**Table 1 nutrients-15-00242-t001:** Characteristics of the studies.

First Author (Year)	Country	Age/Mean/Range	Sample Size Intervention (I), Control (C)	Duration of Intervention
Single Intervention Component: 13 Studies				
*Sensory component approach (single or multisensory): 54%.*				
Caton et al., 2014 [31]	France	4–38 months	I1 = 112 I2 = 112 I3 = 108	5 weeks
de Wild et al., 2016 [33]	Netherlands	2–4 years old	I1 = 26 I2 = 25 I3 = 26 C = 26	8 weeks
Hoppu et al., 2016 [34]	Finland	3–6 years old	I = 44 C = 24	5 weeks
Coulthard and Sealy., 2017 [32]	UK	2–5 years old	I = 21 C1 = 20 C2 = 21	1 day
Nekitsing et al., 2019 [35]	UK	2–5 years old	I1 = 59 I2 = 66 I3 = 65 I4 = 74	15 days
Garcia et al., 2020 [36]	UK	3–5 years old	I = 64 C = 57	4 weeks
Karagiannaki et al., 2021 [37]	Denmark	3–6 years old	I1 = 47 I2 = 32 I3 = 30 C = 50	6 months
2. *Nutrition component approach (anthropometry, food intake, and behaviour): 38%.*				
Alarcon et al., 2003 [47]	Taiwan and the Philippines	3–5 years old	I = 53 C = 51	3 months
Sheng et al., 2014 [48]	China	2.5–5 years old	I = 77 C = 76	4 months
Kim et al., 2015 [46]	Korea	2–5 years old	I = 35 C = 44	2 months (follow-up at 2 months)
Ghosh et al., 2018 [49]	India	2–6 years old	I = 127 C = 128	3 months
Khanna et al. 2021 [50]	India	2–4 years old	I1 = 107 I2 = 107 C = 107	3 months
3. *Parenting component approach: 8%.*				
Sandvik et al., 2019 [39]	Sweden	4–6 years old	I = 65 C = 65	14–16 weeks (follow-up at 12 months)
**Multi-Component Intervention: 3 Studies**				
Bellows et al., 2013 [40]	USA	4 years old (4.7 ± 0.4)	I = 143 C = 107	12 weeks (follow-up at 24 months)
Mallan et al., 2015 [41]	Australia	There are 4 data collection sessions: 1st: 4.3 ± 1.0 months 2nd: 13.7 ± 1.3 months 3rd: 24.1 ± 0.7 months 4th: 44.5 ± 3.1 months	I = 174 C = 166	10 months (follow-up at 10 months and 30 months)
Skouteris et al., 2016 [42]	Australia	2–4 years old	I = 104 C = 97	15 months

**Table 2 nutrients-15-00242-t002:** Intervention components and delivery methods used for the intervention program.

Type of Intervention	Author (Year)	Screening Tools for PE	Intervention Components	Implementation	Outcome Finding
Single-component intervention: sensory component approach (single or multisensory): 54%	Caton et al., 2014 [31]	None (evaluation of intervention using CEBQ)	*Sensory (taste).* Repeated exposure and flavour masking.	Consumption of three types of artichoke: (a) artichoke puree, (b) flavour-learning: artichoke puree and sweetness, and (c) flavour-nutrient learning: artichoke puree and energy. Each subject received 5–10 exposures.	Behaviour assessment: Repeated exposure successfully increased the acceptance of novel food, especially among younger children.
	de Wild et al., 2016 [33]	None (evaluation of intervention using CFNS)	*Sensory (taste and texture)*. Repeated exposure and flavour masking.	Consumption of three types of spinach: (a) pure spinach, (b) cream spinach (flavour masking), and (c) ravioli spinach (hidden). It is frozen and sent home.	Behaviour assessment: All groups increased spinach intake pre-and post-test (even the control group). Offering more vegetables increased vegetable consumption. Ravioli spinach was the least liked. Parents had to chop spinach which may have altered the outcome. Low spinach intake correlated with high levels of food neophobia.
	Hoppu et al. 2016 [34]	None	*Sensory (multisensory).* Multisensory play.	Five sessions of sensory learning (once per week), 20–30 min. Food play (multisensory play) involved a visual card and acting (fairy tales and puppets).	Behaviour assessment: A significant increase in the willingness to try all foods. No change in the control group. In terms of food consumption, the willingness to try carrots, swede, and bilberries was the only significant increase.
	Coulthard and Sealy. 2017 [32]	None (CFNS was used as an outcome)	*Sensory (multiple sensory).* Multisensory play.	This study used visual food play. In the intervention, participants created photos using fruits and vegetables (FV). They were allowed to chop, reshape, or squish the food.	Behaviour assessment: Children in the intervention group tasted more FV (features and not features in the intervention) than in the control groups.
	Nekitsing et al. 2019 [35]	None	*Sensory (multisensory).* Multisensory play and storybooks.	*Multisensory play*: The exercises covered sound (listening to the vegetable’s name and tapping it to hear a sound), sight, touch, and smell (picking and sniffing the various shapes), but not taste. *Storybook*: Storybook featuring targeted vegetables (picture). Throughout the intervention, multiple storybook readings were conducted.	Behaviour assessment: The congruent storytelling and congruent sensory play boosted targeted vegetable consumption, but not the quantity. Sensory exclusively enhanced vegetable consumption.
	Garcia et al. 2020 [36]	Yes (reports by caregivers)	*Sensory (taste and texture).* Preparing a meal together and repetitive vegetable exposure and sensory play.	Parent-child cooking session. Children used cooking tools to learn to chop, grate, mix, and measure food ingredients that included vegetables. Each weekly session encouraged the use of smell.	Behaviour assessment: PE improved at the end of the session. The intervention group scored higher on raw and cooked vegetable tests than the control group. The intervention group had higher individual scores.
	Karagiannaki et al., 2021 [37]	None (evaluation of intervention using CFNS and CEBQ)	Sensory (taste). Repeated exposure.	All group interventions were given 100 g of daikon. The difference was the time exposure (once per week, twice per week, or bi-weekly), but each group received a total of seven-time exposure.	Behaviour assessment: Changes in acceptance based on exposure. Once-a-week exposure and twice-a-week exposure, increase the intake of daikon. The intake peaked at the fourth exposure, then plateauws and dropped by the seventh exposure. CFNS and CEBQ were not used in the result because of the low response rate.
Single-component intervention: nutrition component approach (anthropometry, food intake and behaviour): 38%	Alarcon et al., 2003 [47]	Yes (reports by caregivers and analysis from 3-day diet recall)	*Nutrition (anthropometry)*. Oral nutrition supplement (ONS) and diet counselling (DC).	*ONS*: PediaSure consumed 40 mL/kg/day of the supplement in addition to the irregular diet. *DC:* Individualised counselling by a doctor focusing on healthy eating, portion control, reducing sugar and fat intake, and giving praise when eating refused dishes.	Nutrition status and food intake: Overall, weight, height, weight-for-age, and height-for-age improved in the intervention group compared to the control group.
	Sheng et al., 2014 [48]	Yes (reports by caregivers)	*Nutrition (anthropometry and food intake)*. Oral nutrition supplement (ONS) and diet counselling (DC).	*ONS*: Milk-based powder (S-26 PE GOLD), taken at least 230 mL servings/day. *DC*: An individualised dietary approach using the Child Nutrition Branch Dietary Guidelines for children. Scheduled meals and snacks, portion sizes, a distraction-free mealtime setting, and providing a mealtime role model.	Nutrition status and food intake: Significantly improved nutrition status (weight-for-height and weight-for-age). The intervention group showed higher total energy and consumed more carbohydrates, protein, and micronutrients than the control group. The intervention group had a higher percentage of RNI change than the control group for total energy, calcium, iron, zinc, and vitamins A, C, D, and E.
	Kim et al., 2015 [46]	None (Evaluation of intervention using a modified version of Carruth’s (2004))	*Nutrition (anthropometry, eating behaviour and food intake)* and ONS.	*Herbal supplementation:* SEC-22 (herbs) was provided orally in the mornings and nights after being boiled.	Nutrition status and food intake: No significant results were observed in nutrition status (anthropometry), eating behaviour, and nutrient intake. Follow-up: intervention group consumed more food than the control group (improved carbohydrate vitamin c and thiamine)
	Ghosh et al., 2018 [49]	Yes (set of a question of PE behaviours, identified as a PE if shows two or more behaviours)	*Nutrition (anthropometry)*. Oral nutrition supplement (ONS) and diet counselling (DC).	*ONS*: PediaSure. Children aged 24 to 48 months consumed one serving (224 mL) of ONS, while those aged 48 to 72 months were provided with two servings (448 mL). *DC:* No details provided.	Nutrition status and food intake: The intervention group had a significant increase in weight and improved nutritional status (weight-for-age and BMI-for-age) compared to the control group.
	Khanna et al., 2021 [50]	Yes (set of a question of PE behaviours, identified as a PE if it shows two or more behaviours)	*Nutrition (anthropometry)*. oral nutrition supplement (ONS) and diet counselling (DC).	*ONS*: PediaSure (ONS1-milk base; ONS2-lactose-free) taking 1–2 servings daily. *DC*: The counselling focused on eating a well-balanced diet that included foods from various food categories, improving the diet’s quality, and meeting the child’s daily nutritional needs.	Nutrition status and food intake: Overall, weight-for-height improved significantly in the intervention group (ONS1 and ONS2) compared to the control group. Weight-for-age improved in intervention group 1 but only improved at day 90 for intervention group 2 compared to the control group. Weight improvement in intervention group 1 and intervention group 2 (days 1–30 and day 90) compared to the control group.
Single-component intervention: parenting component approach: 8%	Sandvik et al., 2019 [39]	None (evaluation of intervention using CEBQ and the LBC Lifestyle Behaviour Checklist)	*Parenting*. To support a healthy environment, healthy eating, and physical activity.	Group discussions and practice through role-playing; 10 sessions of creating a suitable environment and parenting support to eat sufficiently were conducted.	Behaviour assessment: PE behaviour did not change in the intervention and control groups throughout the programme. Children with higher baseline PE (CEBQ) lost less weight.
Multiple intervention components	Bellows et al., 2013 [40]	None	*Cognitive Behaviour.* Used vegetable cartoons in activities. *Sensory (taste and texture).* Repeated exposure. *Social and Environmental.* A repeated message from earlier phases.	*Cognitive Behaviour*. Fun and creative activities such as a puppet show, fruit and vegetable mystery bag, a tasting party, and puzzles. *Sensory*-Jicama was offered and repeatedly exposed to the children. Used posters and banners to display in the school environment. Parents received a newsletter (chef’s hat, spatula, and recipe book).	Behaviour assessment: The intervention group preferred targeted vegetables more than the control group. Children who rated the targeted vegetable as “yummy” or “okay” increased their vegetable consumption at the end of the intervention session. The control group children did not necessarily eat more vegetables.
	Mallan et al., 2015 [41]	None (evaluation of intervention using CEBQ)	*Nutrition (food intake)*. Texture and taste variety, and neutral exposure to healthful meals. *Social and Environmental*. Positive feeding environment and toddler eating behaviour management. *Parenting*. Authoritative parenting practice.	Group-interactive sessions. Workbooks were distributed to ensure optimal intervention dose, monitored home-based tactics, and promoted retention.	Behaviour assessment: More fruits, vegetables, and noncore (nutrient-poor, high in saturated fat, and rich in sugar or added salt) items given at 14 months were well accepted at 3.7 years old. A lower fussiness score at 3.7 years related to more vegetables consumed at 14 months. Nutrition status and food intake: 3. There is no association between a child’s BMI z-score at the age of 3.7 years and the number of fruits, vegetables, and noncore foods consumed at 14 months old.
	Skouteris et al., 2016 [42]	None (evaluation of intervention using CFNS and CEBQ)	*Nutrition (food intake and behaviour).* Healthy eating and cooking together. *Parenting.* Parenting behavioural model.	All interventions were organised in a workshop. Discussion and presentation, child play, and healthy food demonstration.	Behaviour assessment: Children in the intervention group consumed more vegetables and fewer snacks, and were more responsive to satiety cues than those in the control group. At twelve months post-intervention, neophobia was lower in the intervention group than in the control group.

## Data Availability

Not applicable.

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
