# Peer review of "Interventions for Picky Eaters among Typically Developed Children—A Scoping Review"

_nutrients, 2023, doi:10.3390/nu15010242_

Round 1

Reviewer 1 Report

The current manuscript reviewed studies on interventions for typically developing children with picky eating behavior. Below are my suggestions: 

1. In Figure 1, please straighten the arrow pointing to "Records screened" and "Studies included in review". 

2. The context in Figure 2 is difficult to tell. Please replace it will a clearer one.

3.  Table 2 looks very busy and confusing. Please re-edit the layout of it.

4. From the initial 1294 papers, large numbers of papers were excluded. There are 250 papers excluded for subjects not typically developing preschool children. So what is the rationale for paying attention to the typically developing picky-eating children in the current review?

5. The conclusion of this manuscript is not very clear.

Reviewer 2 Report

Comments:

Review of the paper

nutrients- 2068336

This article aims to study the interventions for picky eaters among typically developed children. The paper investigated the contents of three databases, PubMed, Emerald Insight, and Web of Science, counted all the articles in the three databases about child picky eating, and did not find which intervention showed the most obvious improvement in child picky eating. The survey content is more, but they are the content explored by predecessors, but this survey is only part of the content statistics, the innovation degree is too low, and the conclusion is simple. For the improvement of this manuscript, some comments are listed as follows.

ABSTRACT

-The multiple measurements of the outcomes were insufficient to draw conclusions about which type of intervention was most effective in addressing physical education in typically developing children. The main results and methods of this study are not well summarized and the description of the results is ambiguous; the authors should include more results and methods in their summary and make clear conclusions.

MATERIALS AND METHODS

- This survey only investigated the relevant articles in the three databases, and did not do the relevant questionnaires. The data relied on the previous research, and even only counted the results of some mainstream intervention methods. The research depth and innovation of the experiment were not enough. Please provide quality rating and exclusion criteria for the included literature as support.

RESULTS

-The results of the article are simple and unclear, and it is impossible to determine which intervention is the best way to improve children 's picky eating. It is recommended to summarize further for accurate conclusions.

DISCUSSION

-The data of the article are related articles retrieved from three databases. The research content of these articles is carried out in the form of offline questionnaires. Most of the questionnaires are filled out according to the children 's parents ' understanding of the children 's diet. Parents ' comments are somewhat subjective and certainly influenced by geographical scope. So, the accuracy of the data is not guaranteed.

REFERENCES

- The article is studied by investigating the relevant literature in the database, which is highly cited and deserves praise. But the subject of the study of the article is children, and there are 4-year-old infants and young children in the literature, which should not be. Age-appropriate groups are recommended.

Reviewer 3 Report

Nutrition in picky children is a very specific branch of pediatric diabetology. There are several dietary interventions well described in the study, but a clear assessment of which of the strategies is the most effective is lacking, which is given by the individual specific response to the intervention of the child. The topicality of the topic and its diversity is also reflected in the number of evaluated studies that meet the inclusion criteria, the number of which is approximately 1% of the assessed set. In order to evaluate the effectiveness of a PE intervention, a clear criterion of effect must be established in order to determine which of the interventions is the most effective. Unfortunately, this effectiveness is affected by a number of factors, many of which are mentioned in the study, but others are missing. E.g. how children's PE is reflected in their weight status, economic demand, qualitative structure of PE, energy demand, etc. All this should be mentioned and evaluated in the discussion. Where at the same time I lack clear limits of this study. Similarly, the conclusion deserves more concreteness towards practical recommendations.

Round 2

Reviewer 1 Report

The authors solved my concerns in the revision. I accept the present form for publishing.

Reviewer 3 Report

A well-developed topic, where I recommend shortening the conclusion somewhat and also making it more concrete.